# Design of Polarization-Independent and Wide-Angle Broadband Absorbers for Highly Efficient Reflective Structural Color Filters

**DOI:** 10.3390/ma12071050

**Published:** 2019-03-30

**Authors:** Kyu-Tae Lee, Daeshik Kang, Hui Joon Park, Dong Hyuk Park, Seungyong Han

**Affiliations:** 1Department of Physics, Inha University, Incheon 22212, Korea; ktlee@inha.ac.kr; 2Department of Mechanical Engineering, Ajou University, San 5, Woncheon-Dong, Yeongtong-Gu, Suwon 16499, Korea; dskang@ajou.ac.kr; 3Department of Energy Systems Research, Ajou University, San 5, Woncheon-Dong, Yeongtong-Gu, Suwon 16499, Korea; 4Department of Electrical and Computer Engineering, Ajou University, San 5, Woncheon-Dong, Yeongtong-Gu, Suwon 16499, Korea; 5Department of Chemical Engineering, Inha University, Incheon 22212, Korea

**Keywords:** absorbers, structural colors, resonators, Fabry–Pérot

## Abstract

We propose a design of angle-insensitive and polarization-independent reflective color filters with high efficiency (>80%) based on broad resonance in a Fabry–Pérot cavity where asymmetric metal-dielectric-metal planar structures are employed. Broadband absorption properties allow the resonance in the visible range to remain nearly constant over a broad range of incident angles of up to 40° for both s- and p-polarizations. Effects of the angles of incidence and polarization state of incident light on the purity of the resulting colors are examined on the CIE 1931 chromaticity diagram. In addition, higher-order resonances of the proposed color filters and their electric field distributions are investigated for improved color purity. Lastly, the spectral properties of the proposed structures with different metallic layers are studied. The simple strategy described in this work could be adopted in a variety of research areas, such as color decoration devices, microscopy, and colorimetric sensors.

## 1. Introduction

Color filters, which transmit or reflect wavelengths in the visible portion of the spectrum, have played a vital role in a wide variety of fields such as displays, decorations, imaging sensors, anti-counterfeiting and light-emitting devices. Conventional color filters, however, have used chemical pigments that are highly sensitive to constant ultraviolet (UV) light illumination, many chemicals, moisture and high temperature, all of which can cause a gradual degradation of their performance over time [1]. In order to address these challenges, many researchers have proposed and experimentally demonstrated various nanostructures that exploit optical interference effects in a thin-film configuration [2,3,4,5,6], photonic crystals [7,8,9], guided-mode resonances (GMRs) in subwavelength grating waveguides [10,11,12,13] and surface plasmon resonances (SPRs) in nanocavities patterned at the subwavelength scale [14,15,16,17,18,19,20,21,22,23,24,25,26,27]. These colors, called structural colors, have attracted substantial attention for their potential in creating distinctive colors with capabilities of achieving much improved efficiency, slim dimension, long-lasting stability and great scalability. Moreover, such structural color filters typically employ nanostructures with an ultrathin thickness, thereby providing distinct possibilities for realizing flexible display systems, reducing material usage and simplifying fabrication processes. Although numerous nanostructures have been studied intensively in recent years for color generation, there have been great difficulties in retaining highly efficient optical properties over a wide angle of incidence. Various strategies that rely on a localized resonance in patterned nanostructures [28,29], a phase compensation [30,31,32], a small refraction in photonic crystals [33,34] and a strong interference effect in nanocavities [35,36,37,38,39] have been reported in order to mitigate angle-dependent properties. However, many complicated fabrication steps need to be involved to create the nanostructures with the subwavelength patterns, which limit their large-scale applications. Most of the materials with high index of refraction also have a large absorption coefficient, thus causing a significant reduction in color efficiency. In addition to the angle-sensitive performance, a variety of the structural colors in previous reports use one-dimensional (1D) periodic grating structures that only work for a certain polarization state of incident light. Therefore, there is a crucial need for developing a new approach with novel functionalities that can resolve the aforementioned issues at the same time.

In this work, a scheme of wide-angle, polarization-independent and high-efficiency reflective structural color filters that exploit a broad optical resonance in an asymmetric planar cavity configuration is proposed. Broad spectral properties lead to the angle invariant performance, keeping highly-efficient colors over a broad angle of incidence up to 40° regardless of polarization state of incident light. Moreover, the dependence of optical properties and their resulting colors on the incident angles, polarizations and higher-order resonances is rigorously investigated, which is studied on the CIE chromaticity diagram. Furthermore, effects of top and bottom metallic mirrors on the optical properties are explored. The proposed approach holds promise for diverse applications, including microscopy, color-selective optoelectronics and decorative solar panels.

## 2. Results and Discussion

Figure 1a depicts the schematic diagram of the proposed broadband absorber-based highly efficient reflective structural color filters with angle-insensitive and polarization-independent performances. The cavity structure simply consists of a dielectric layer sandwiched by two different metallic layers with anti-reflective (AR) coatings atop in an asymmetric configuration. Titanium dioxide (TiO_2_) is a wide bandgap material whose optical absorption losses in the visible wavelength range are nearly negligible, which is used as a cavity medium. A change in the thickness of TiO_2_ leads to shifted resonances. Silicon dioxide (SiO_2_), which is a lossless dielectric material, is employed as the AR layer that mitigates the reflection at the top surface of the metallic mirror. Chromium (Cr) is used as a mirror with relatively constant and lower reflectivity, implying that it uniformly absorbs the visible range. Such optical absorptions of Cr are required to attain the broadband absorption property and thus reflective color generation with high efficiency. Aluminum (Al) is utilized for providing high reflections at a bottom surface. When employing lossy metals at the bottom of the structure, the resonance is too broad to create the colors. Effects of both top and bottom metallic layers on the spectral reflectance properties will be studied in the last section of this paper. It is also important to note that both Cr and Al feature low cost, abundant, high stability and good adhesion to numerous materials, thereby providing opportunities for practical applications. As optically thick metallic mirror is used at the bottom, either silicon (Si) or glass can be used for a substrate. Figure 1b presents measured complex refractive indices of the dielectric materials by using a spectroscopic ellipsometer (Elli-SE, Ellipso Technology Co.), which are used in the simulation. Refractive indices of Cr and Al are retrieved from Johnson and Christy [40]. Figure 2a–c presents 2D contour plots of optical absorptions as a function of the wavelength and the SiO_2_ thickness with fixed thicknesses of Cr and Al at 5 and 100 nm for blue, green and red colors, respectively, which are obtained by using the transfer matrix method. Those plots show the optimized thickness of the AR layer. Since an optically thick metal is used as a substrate, no light is transmitted through the entire structure, which implies that spectral reflectance properties are simply complementary spectra of the optical absorption profiles. In Figure 2a, it is found that the reflection spectrum for the blue color shows not only highly suppressed reflections (i.e., high absorptions) at resonances between 500 nm and 800 nm but high reflection efficiency (i.e., low absorptions) at off-resonance wavelengths between 400 nm and 500 nm along with a narrow bandwidth when choosing the thickness of around 80 nm for the AR coating. We note that capabilities of filtering a narrowband of desired wavelengths and suppressing other wavelengths are essential to achieve high color purity. Similarly, an 80 nm thick AR layer is found to be an adequate thickness for both green and red colors as well, as can be seen from Figure 2b,c. Figure 2d displays the 2D contour plot of optical absorptions as a function of the wavelength and the TiO_2_ thickness. With increasing the thickness of the spacer layer, a resonance (i.e., absorption peak) shifts toward a longer wavelength range and higher-order resonances appear at shorter wavelengths.

Figure 3a–c describes simulated spectral reflectance curves of the proposed structural color filters at normal incidence. The simulations based on the transfer matrix method are carried out to obtain the reflection spectra. As the thick metallic substrate is used in our design, there is no light transmission through the structure (T = 0) so the reflection spectrum can be easily attained from R = 1− A where R and A are the reflection and the absorption, respectively. In the proposed color filter structures consisting of anti-reflective coating (SiO_2_)—lossy mirror (Cr)—cavity medium (TiO_2_)—highly reflecting mirror (Al), a highly lossy metal like Cr is chosen as a top mirror to render the Q-factor of the Fabry–Pérot (F-P) cavity lower so that the broadband absorption resonance is attained. The absorption resonance corresponds to the valley in the reflection. Controlling a bandwidth of the absorption resonance and a free spectral range in a proper manner allows a reflection peak to be created for the reflective RGB color generation. In Figure 3, the reflection peak does not correspond to the F-P resonance, while reflection valleys correspond to the F-P resonance. Thicknesses of a spacer layer, TiO_2_, are 140, 100 and 65 nm for red, green, and blue (RGB) colors, while the thicknesses of SiO_2_, Cr and Al are fixed at 80, 5 and 100 nm, respectively. With increasing the thickness of the spacer layer, the resonance shifts toward the longer wavelength. As can be seen from black solid lines that are obtained from the structure where TiO_2_ on top of Al without the top lossy metallic mirror, a resonance behavior with weak optical absorptions and hence high reflections is observed for all RGB colors despite a broad spectral property. It is difficult to produce desired vivid colors from such flat resonances with high reflections. By putting the metallic film that evenly absorbs the entire visible wavelength range, the resonance effect with strong optical absorptions is achieved (red solid curves), thus creating distinctive reflection colors as compared to the case without the lossy metal. Placing the AR layer on top of the structure with the optimized thickness as shown in Figure 2 allows the resonance to be sharper than the two previous cases as represented by blue solid lines. It is apparent that the efficiency of the reflective RGB colors is around 90%, which is much higher than that of the conventional pigment-based color filters widely used in liquid crystal display technologies. Although both blue and red colors with high efficiency can be produced when employing the AR coating and the lossy metallic mirror, the resonance for the green color to still be still too broad, presenting the degraded color purity. In order to improve the purity of the green color, the cavity thickness increases from 100 nm to 210 nm so that the higher-order F-P resonance can be exploited, which exhibits much sharper spectral properties than the fundamental mode. As is seen from a green curve in Figure 3b, the second-order resonance (absorption peak) occurring at 725 nm is remarkably sharp, and the third-order resonance appears at 425 nm, both of which have high efficiency. This results in a sharp reflection peak at 550 nm with fairly narrow bandwidth as compared to other three cases. We note that exploiting the second-order resonance for the blue color leads to the cyan color because the spectral reflectance curve encompasses the longer wavelength components, while the shorter wavelengths are contained in the reflectance curve for the red color causing the magenta color employing the second-order resonance. Thus, 180 and 250 nm of TiO_2_ are used to exploit the second-order resonance for blue and red colors, respectively.

Next, the purity of the resulting reflective colors is investigated on the CIE 1931 chromaticity diagram as illustrated in Figure 4. As we discussed in Figure 3, although the broad resonance can be achieved from the structure without the top lossy metallic mirror (i.e., TiO_2_-Al), the optical absorptions are insignificant and thus the reflections are too high across the entire visible range. This is because there are no media that strongly absorb the visible wavelength range, which can lead to an undesired indistinct color as can be seen from the chromaticity diagram (black stars). Color coordinates (x, y) calculated from the reflection spectra (black solid lines) studied in Figure 3 are (0.347, 0.327), (0.336, 0.341) and (0.313, 0.324) for the RGB colors, respectively. Putting 5 nm thick Cr on top of the structure (i.e., Cr-TiO_2_-Al) resulted in enhanced color purity as the reflections are markedly reduced at the resonance, while the reflections are slightly diminished at the off-resonances. Such improved color purity can be clearly observed on the chromaticity diagram (black squares) showing that the color spaces move toward an outer part of the diagram. The calculated color coordinates (x, y) for the Cr-TiO_2_-Al structure, represented by red solid lines of reflection spectra in Figure 3, are (0.438, 0.350), (0.341, 0.372) and (0.196, 0.169) for the RGB colors, respectively. With the AR coating atop the structure (i.e., SiO_2_-Cr-TiO_2_-Al), the color purity can be further improved, because of the reflections being additionally suppressed at the off-resonance wavelengths as denoted by black circles. The color spaces (x, y) attained from the reflection spectra (blue solid lines) in Figure 3 are (0.449, 0.313), (0.345, 0.383) and (0.191, 0.138) for the RGB colors, respectively. However, it is obvious that a position of the calculated color coordinates for the green color is still far from a green region on the chromaticity diagram, which is attributed to the broad resonance that encompasses the off-resonance wavelength components. In order to achieve the right green color, it is required to further suppress the reflections at the off-resonance wavelengths, which can be enabled by employing higher-order resonances. This is due to the fact that the higher-order resonances have the higher intensity of electric field at the interface between the dielectric and the metal, thus enabling strong optical absorptions in the lossy metallic film (Cr), as presented in Figure 5. Figure 5a,b shows normalized intensity distributions of electric field into the green colored structures that exploit the fundamental resonance mode and the higher-order resonances, respectively, which are attained by using the transfer matrix method. With 100 nm thick TiO_2_ layer as exhibited in Figure 3b, the first- and second-order resonances appear at 800 (red color) and 400 nm (blue color), which can be verified by the field profile displayed in Figure 5a. By employing the higher-order resonances as depicted in Figure 5b, the second- and third-order resonances occur at 700 (red color) and 425 nm (blue color), both of which show higher field intensity in the Cr film, thus yielding much suppressed reflections at the resonances. Such highly reduced reflections are responsible for achieving distinctive green color. In addition to the strong absorptions, the third-order resonance appears at 425 nm, thereby rendering the reflection peak at 550 nm with the narrow spectral band. This leads to remarkably improved purity for the green color as can be observed from Figure 4 (black triangles). It is noted that the purity of both blue and red colors gets worse with the higher-order resonance as the longer and shorter wavelengths are also reflected as revealed in Figure 3. The resulting colors are found to be cyan and magenta for blue and red, respectively. The color coordinates for the case of employing the higher-order resonances (green solid lines in Figure 3) are (0.395, 0.241), (0.317, 0.479) and (0.145, 0.177) for the RGB colors, respectively.

Figure 6 depicts 2D contour plots of the reflections as a function of wavelength and angles of incidence for (a)–(c) s- and (d)–(f) p-polarizations, respectively, which are calculated by the transfer matrix method. It is apparent that the reflection peaks represented by the red color are almost insensitive with respect to the angles of incidence up to 40° for both s- and p-polarizations. This is because the broadband spectral properties, which are relatively insensitive to the changes in the thickness, the incident angle, and the polarization state, are exploited to create the resonance at visible frequencies and hence the reflection colors. The high refractive index of the cavity medium (i.e., TiO2) is also responsible for achieving the angle- and polarization-insensitive performances. Moreover, the highly efficient reflections are maintained over a broad range of the incident angles, which are highly desired in diverse applications.

Figure 7 presents color spaces calculated from the reflection spectrum at different oblique angles of incidence ranging from 0° to 40° for both s- and p-polarizations described on the CIE 1931 chromaticity diagram. As observed from the angle-resolved reflection spectra exhibited in Figure 6, the positions of the reflection peak remain nearly constant over a wide angle of incidence without sacrificing the reflection efficiency, indicating that there is little color variation with changing incident angles and polarizations. Such angle- and polarization-insensitive properties can also be validated by investigating the calculated color spaces as illustrated in Figure 7. It is clear that the calculated color coordinates for different angles and polarizations are located at similar regions for the individual RGB colors, which are summarized in Table 1.

Lastly, effects of both top and bottom metallic mirrors on the spectral reflectance properties are studied. Figure 8a–c reveals calculated spectral reflectance curves for different top metals with the same thickness of 5 nm in the structure at normal incidence, which are obtained by the transfer matrix method. The thicknesses of SiO_2_, TiO_2_ and Al layers are the same with those in Figure 3. With metals such as silver (Ag), gold (Au) and copper (Cu), it is difficult to see the strong optical resonance effect as the reflection from the top surface seems to be too low due to a very thin film thickness, generally exhibiting a flat spectral reflectance. Although the resonances appear with the Al top mirror, the spectral response is not broad enough to create the individual colors with high purity. Other metals like nickel (Ni), titanium (Ti) and tungsten (W) are also not a good candidate as the top mirror, which could be due to their relatively lower absorptions across the entire visible wavelength range as compared to Cr. It is noted that the positions of the resonant wavelength are slightly shifted with different metallic mirrors, which could be credited to the fact that the metals have their own penetration depth, thus leading to varied effective optical path length [37]. In Figure 8d–f, calculated reflection spectra with different bottom metallic mirrors at normal incidence are described. In this case, novel metals like Ag and Au can also function as a highly efficient metallic substrate instead of Al. Due to the lowest absorption losses of Ag in the visible regime among noble metals, the efficiency at the resonance is higher than that obtained with Al. However, Al is selected in our design due to its unique properties such as natural abundance, low-cost, good adhesion and high stability, thus making it more practical for diverse applications. When employing other lossy metals like Ni, Ti and W as the substrate, the absorption losses in the visible range are too significant, thereby resulting in a flat spectral reflectance with very low efficiency. From this study, it is demonstrated that both Cr and Al would be the good choice as the top and bottom metallic mirrors to practically accomplish highly efficient reflective structural color filters. As the only deposition methods without complicated and expensive fabrication processes can be involved to experimentally demonstrate the proposed structure, it is thus expected to easily create such schemes on a flexible substrate, which can offer the fascinating possibility of realizing various applications on a flexible platform.

## 3. Conclusions

In conclusion, we have shown a scheme of creating highly efficient (>80%) reflective RGB colors with angle-insensitive and polarization-independent properties exploiting broad optical resonances in ultrathin planar nanostructures simply consisting of four layers. The resonant wavelengths of the proposed color filter structures remain nearly constant over a broad range of incident angles for both s- and p-polarizations, which are ascribed to the broad optical resonance effects. Besides, it is shown how both incident angles and polarizations have influence upon the reflection spectrum and hence the colors produced by the nanostructures. It is also demonstrated that improving color purity for a certain color is enabled by employing higher-order resonances. Furthermore, the dependence of the spectral reflection properties on different metallic layers is examined. The approach proposed here potentially paves the way towards numerous applications such as colored solar cells, color decorations, and wavelength-selective photodetectors.

## Figures and Tables

**Figure 1 materials-12-01050-f001:**
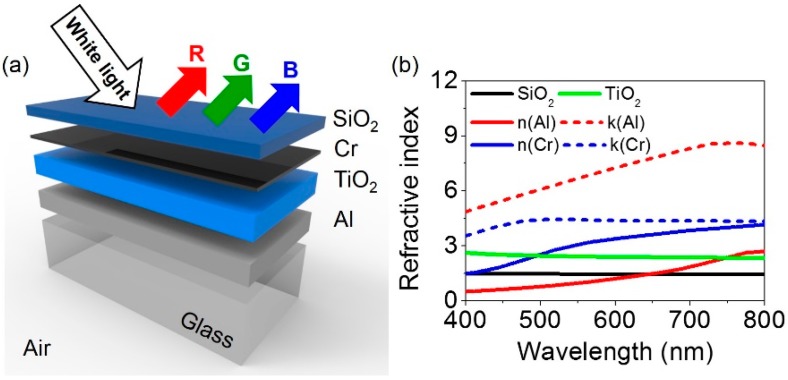
(**a**) Schematic diagram of the proposed nanostructures that can absorb a broad wavelength range of visible light to produce reflective colors that are insensitive with respect to both incident angles and polarizations. The structure is simply comprised of alternating multiple layers of dielectric and metallic films with different combinations on a glass substrate. (**b**) Refractive indices of materials used in the simulations.

**Figure 2 materials-12-01050-f002:**
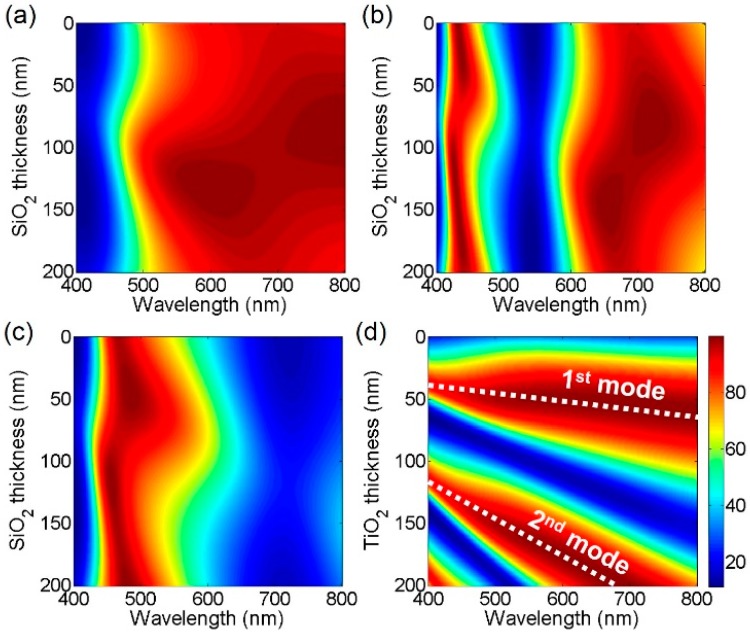
2D contour plots of optical absorptions as a function of a wavelength and a SiO_2_ layer thickness for (**a**) blue, (**b**) green and (**c**) red colors. (**d**) 2D contour map of absorptions as a function of a wavelength and a thickness of TiO_2_.

**Figure 3 materials-12-01050-f003:**
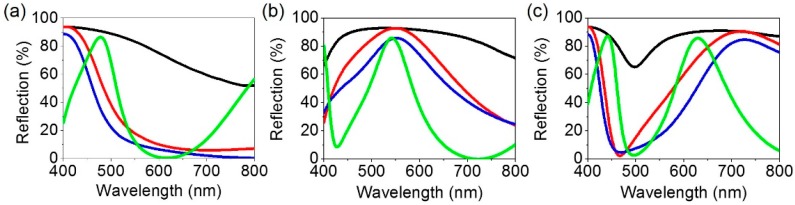
Simulated spectral reflectance curves at normal incidence for (**a**) blue, (**b**) green and (**c**) red colored structures. Black, red, blue and green solid lines represent the reflection spectrum obtained from the structures consisting of TiO_2_ (B: 65 nm, G: 100 nm, R: 140 nm)/Al (100 nm), Cr (5 nm)/TiO_2_ (B: 65 nm, G: 100 nm, R: 140 nm)/Al (100 nm), SiO_2_ (80 nm)/Cr (5 nm)/TiO_2_ (B: 65 nm, G: 100 nm, R: 140 nm)/Al (100 nm) and SiO_2_ (80 nm)/Cr (5 nm)/TiO_2_ (B: 180 nm, G: 210 nm, R: 250 nm)/Al (100 nm), respectively.

**Figure 4 materials-12-01050-f004:**
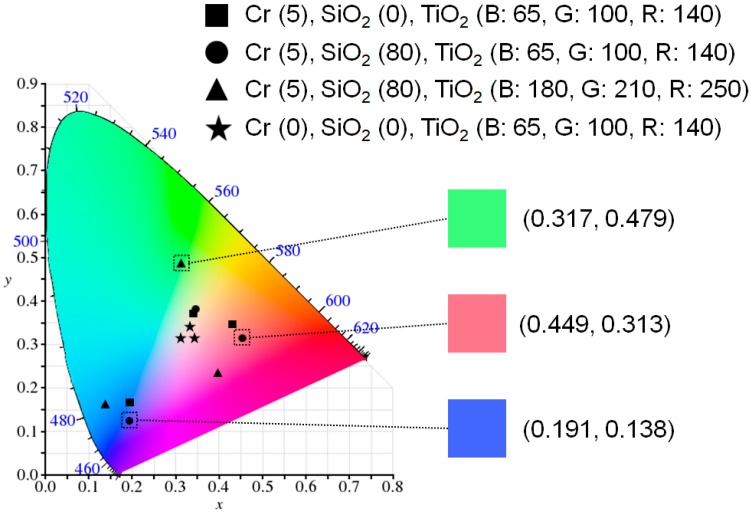
A representation of color coordinates calculated from the reflection spectra studied in Figure 3 illustrated on the CIE 1931 chromaticity diagram.

**Figure 5 materials-12-01050-f005:**
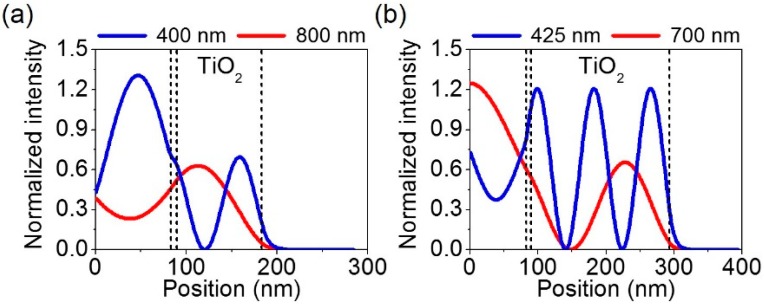
Normalized intensity distributions of electric field into the proposed structures where the thickness of TiO_2_ layer is (**a**) 100 nm and (**b**) 210 nm with fixed thicknesses of SiO_2_, Cr and Al at 80, 5 and 100 nm.

**Figure 6 materials-12-01050-f006:**
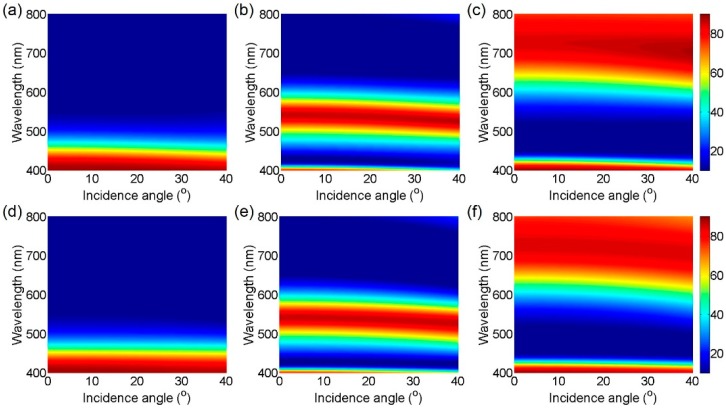
Simulated 2D contour plots of reflection as a function of a wavelength and an angle of incidence displaying that the reflection peaks are insensitive to incident angles up to 40° for (**a**–**c**) s- and (**d**–**f**) p-polarizations. The broad optical resonance effects result in both polarization-independent and angle-invariant characteristics. Red and blue colors represent high and low reflection efficiency in color maps.

**Figure 7 materials-12-01050-f007:**
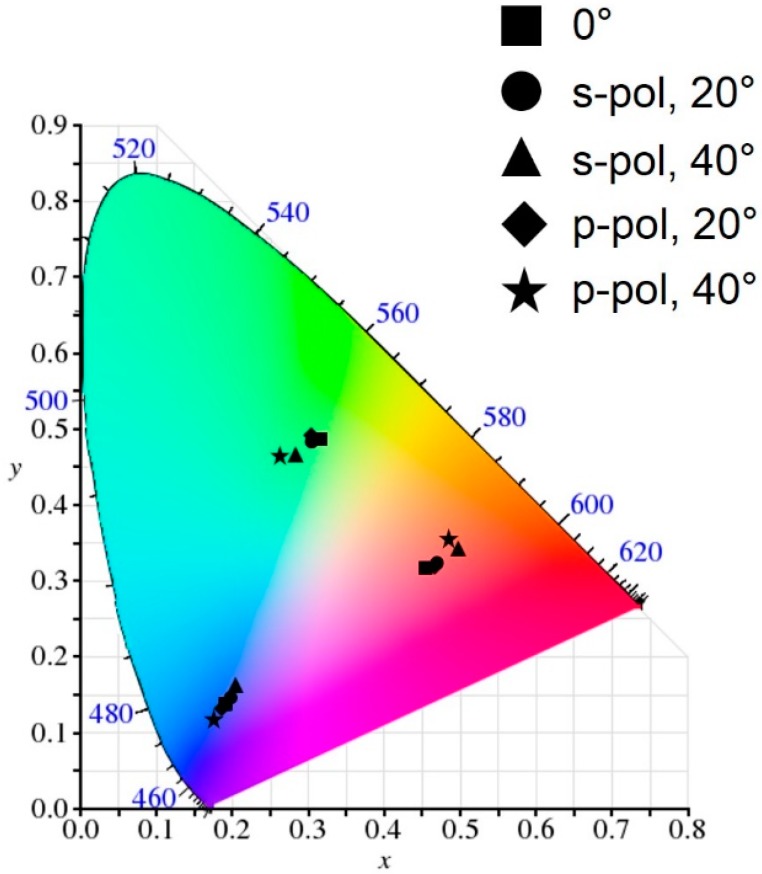
An illustration of color coordinates attained from the angle-resolved reflection spectra in Figure 6 for several oblique angles of incidence under s- and p-polarized light illumination, which is described on the CIE 1931 chromaticity diagram.

**Figure 8 materials-12-01050-f008:**
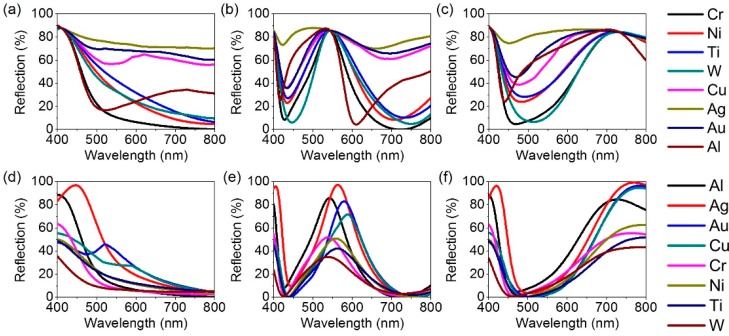
Calculated reflection spectra with different (**a**–**c**) top and (**d**–**f**) bottom metallic layers at normal incidence for blue, green and red colors.

**Table 1 materials-12-01050-t001:** Summary of color coordinates for different angles of incidence (0°, 20° and 40°) and different polarizations (s- and p-polarized light).

Polarization		s-Polarization		p-Polarization	
Angle of Incidence	0°	20°	40°	20°	40°
**Blue**	(0.191, 0.138)	(0.194, 0.141)	(0.206, 0.158)	(0.189, 0.137)	(0.176, 0.121)
**Green**	(0.317, 0.479)	(0.304, 0.475)	(0.276, 0.460)	(0.303, 0.477)	(0.259, 0.461)
**Red**	(0.449, 0.313)	(0.463, 0.326)	(0.491, 0.346)	(0.459, 0.327)	(0.478, 0.366)

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
