# Peer review of "Design of Polarization-Independent and Wide-Angle Broadband Absorbers for Highly Efficient Reflective Structural Color Filters"

_materials, 2019, doi:10.3390/ma12071050_

Round 1
Reviewer 1 Report
The paper is well written, with clear objectives. The topic is an important subject.
Author Response
Reviewer #1
Summary: “The paper is well written, with clear objectives. The topic is an important subject.”
Our response: We thank the reviewer for the positive remark.

Reviewer 2 Report
The authors report angle-insensitive color filtering devices working under reflection mode. Overall, the topic is interesting and may draw great attention so it is suitable for Materials. However, the following issues must be fully addressed: 1. As a scientific paper reporting color filters, the whole manuscript does not include a clear image showing the images of filtered output colors. The CIE-1931-chromaticity-diagram-only results are not convincing; 2. Fig.1 (b) presents measured complex refractive indices of the dielectric materials. Do the authors have any experimental (measured) results showing the sample surface and cross-section of the layered device (SEM images) and any experimental results (measured) of output colors? 3. Did the authors consider carrying out experimental fabrication of this color filter and measure the real efficiency? They claim 80% which is simulation results. What real efficiency are we expecting experimentally? 4. The introduction part needs to be improved. References are not cited in a smooth way (Ref. 2-18 and 19-30); 5. Some important refs are missing so please add them in (e.g. Optics Express Vol. 26, Issue 19, pp. 25178-25187; Nanoscale 5 (14), 6243-6248; Advanced Materials 24 (23), OP131-OP135; Materials 7 (2), 1296-1317; Applied Physics Letters 99 (3), 033105; Applied Physics A 107 (1), 49-54; Nanotechnology 25 (45), 455203; Materials 10 (8), 944; Materials & Design 96, 64-67).
Author Response
Reviewer #2
Comment #1: “The authors report angle-insensitive color filtering devices working under reflection mode. Overall, the topic is interesting and may draw great attention so it is suitable for Materials. However, the following issues must be fully addressed: As a scientific paper reporting color filters, the whole manuscript does not include a clear image showing the images of filtered output colors. The CIE-1931-chromaticity-diagram-only results are not convincing.”
Our response: We thank the reviewer for this suggestion.
In revisions, images and the corresponding color coordinates of the reflective RGB colors calculated from the reflection spectra were added to Figure 4.
Comment #2: “Fig.1 (b) presents measured complex refractive indices of the dielectric materials. Do the authors have any experimental (measured) results showing the sample surface and cross-section of the layered device (SEM images) and any experimental results (measured) of output colors?”
Our response: As presented in Figure 1 (b), refractive indices of both SiO2 and TiO2 were measured and then used in the simulation to design the structural color filters and to investigate their optical properties. We initially tried to fabricate the devices using an e-beam evaporation, but we realized that oxygen gas needs to flow into the chamber when deposited TiO2 using the e-beam evaporator to obtain TiO2. Otherwise, TiOx was deposited, which presented a lower transmission than TiO2. However, our e-beam evaporator is not equipped with such a gas flow system. We also wanted to try to deposit zinc sulfide (ZnS) that has a high refractive index with a negligible extinction coefficient at visible frequencies instead of TiO2. However, the e-beam evaporator staff didn’t allow us to deposit ZnS due to the possibility of the chamber contamination. As the evaporator is a common facility, we didn’t find other options so we decided to complete the work by the simulation.
Comment #3: “Did the authors consider carrying out experimental fabrication of this color filter and measure the real efficiency? They claim 80% which is simulation results. What real efficiency are we expecting experimentally?”
Our response: We thank the reviewer for highlighting this point. However, we decided to finish the work by the simulation as explained in our response to comment #2.
Comment #4: “The introduction part needs to be improved. References are not cited in a smooth way (Ref. 2-18 and 19-30); 5. Some important refs are missing so please add them in (e.g. Optics Express Vol. 26, Issue 19, pp. 25178-25187; Nanoscale 5 (14), 6243-6248; Advanced Materials 24 (23), OP131-OP135; Materials 7 (2), 1296-1317; Applied Physics Letters 99 (3), 033105; Applied Physics A 107 (1), 49-54; Nanotechnology 25 (45), 455203; Materials 10 (8), 944; Materials & Design 96, 64-67).”
Our response: We thank the reviewer for this suggestion.
In revisions, the introduction was modified. In addition, the references were rearranged, and the papers suggested by the reviewer were cited.

Reviewer 3 Report
Please check the attached review letter.

Author Response
Reviewer #3
Summary: “The paper has technically sound in terms of analyzing the proposed Fabry-Perot structure in terms of its reflectance at specific wavelengths. The selection of materials such as dielectric layers and metal layers are well supported by the analytical solutions. However, I have some comments on the paper.”
Our response: We thank the reviewer for the positive remark.
Comment #1: “It will be good to add some descriptions with equations on the method of getting the presented result for casual readers. It seems discussion about the calculation of the result is not very well described in the paper.”
Our response: We thank the reviewer for this suggestion.
In revisions, we added more descriptions associated with the working principle of the proposed color filter structure: “In the proposed color filter structures consisting of anti-reflective coating (SiO2) – lossy mirror (Cr) – cavity medium (TiO2) – highly reflecting mirror (Al), a highly lossy metal like Cr is chosen as a top mirror to render the Q-factor of the Fabry–Pérot (F-P) cavity lower so that the broadband absorption resonance is attained. The absorption resonance corresponds to the valley in the reflection. Controlling a bandwidth of the absorption resonance and a free spectral range in a proper manner allows a reflection peak to be created for the reflective RGB color generation. In Figure 3, the reflection peak doesn’t correspond to the F-P resonance, while reflection valleys correspond to the F-P resonance.”
Comment #2: “About application study (page 2, line 62~63) – How the proposed device can be applied to display technologies, light-emitting diodes, and imaging sensors? First, the proposed design is a reflective device which has a full mirror on one side of the Fabry-Perot mirrors. If both mirrors are partial mirrors, the presented advantages of the device are still valid? If those are, please add some supporting analysis or arguments in the paper. Secondly, the color selection of the device is performed by changing the thickness of the TiO2 layer. In order to use the device for display or imaging sensors, there must be different thicknesses at each pixel. How can multiple thicknesses be achieved with high precision at different locations? In my opinion, to make the statement of the authors on applications strong, these two issues should be addressed properly.”
Our response: We thank the reviewer for highlighting this point. We agree with the reviewer that our approach would not be suitable for display and imaging applications because the thickness of the cavity medium needs to be varied to tune the colors although the individual color pixel can be created by three separate lithographic processes. Although we also tried to employ both optically thin mirrors in the structural color filter devices to see if the transmissive RGB colors can be achieved, the transmission efficiency was found to be pretty low because a highly lossy metal (Cr) is used to exploit the broadband resonance behavior. Thus, we believe that it is necessary to modify the introduction regarding the application study.
In revisions, we removed the unsuitable applications such as display, printing, and imaging pointed out by the reviewer, and included other potential applications such as color decoration devices, microscopy, wavelength-selective photodetectors, and colored solar cells in the abstract, introduction, and conclusion.

Reviewer 4 Report
This manuscript needs a revision:
Authors should describe simulation methods: software used, resolution,…
The mechanism for the polarization-independent and wide-angle absorption needs to be discussed.
Fabry–Pérot resonance is mentioned in the abstract but is related to the simulation results. Fig. 3 shows a TiO2 thickness-dependent absorption. Is it a F-P effect?
The electric field is mentioned in the abstract. The e-field intensity needs to be included in the manuscript.
Author Response
Reviewer #4
Comment #1: “This manuscript needs a revision: Authors should describe simulation methods: software used, resolution,…”
Our response: We thank the reviewer for this suggestion.
In revisions, we added the simulation method based on the transfer matrix method to the descriptions related to Figures 2, 3, 5, 6, and 8.
Comment #2: “The mechanism for the polarization-independent and wide-angle absorption needs to be discussed.”
Our response: We thank the reviewer for highlighting this point.
In revisions, we modified and added the following statements: “It is apparent that the reflection peaks represented by the red color are almost insensitive with respect to the angles of incidence up to 40° for both s- and p-polarizations. This is because the broadband spectral properties, which are relatively insensitive to the changes in the thickness, and the incident angles, and the polarization state, are exploited to create the resonance at visible frequencies and hence the reflection colors. The high refractive index of the cavity medium (i.e., TiO2) is also responsible for achieving the angle- and polarization-insensitive performances.”
Comment #3: “Fabry–Pérot resonance is mentioned in the abstract but is related to the simulation results. Fig. 3 shows a TiO2 thickness-dependent absorption. Is it a F-P effect?”
Our response: We thank the reviewer for pointing out this. We believe that the proposed structure relies on a broadband F-P resonance consisting of anti-reflective coating (SiO2) – lossy mirror (Cr) – cavity medium (TiO2) – highly reflecting mirror (Al). Due to the high absorption of the top mirror (Cr), the quality-factor (Q-factor) of the F-P cavity is pretty low, leading to the broadband absorption resonance. The absorption resonance corresponds to the valley in the reflection. By controlling a bandwidth of the absorption resonance and a free spectral range properly, we are able to create a reflection peak to produce the reflective RGB colors. In Figure 3, the reflection peak doesn’t correspond to the F-P resonance, while reflection valleys correspond to the F-P resonance. We believe that adding a couple of descriptions regarding the F-P resonance in the proposed color filter structures to the manuscript is beneficial to the readers.
In revisions, we added the following statements: “In the proposed color filter structures consisting of anti-reflective coating (SiO2) – lossy mirror (Cr) – cavity medium (TiO2) – highly reflecting mirror (Al), a highly lossy metal like Cr is chosen as a top mirror to render the Q-factor of the Fabry–Pérot (F-P) cavity lower so that the broadband absorption resonance is attained. The absorption resonance corresponds to the valley in the reflection. Controlling a bandwidth of the absorption resonance and a free spectral range in a proper manner allows a reflection peak to be created for the reflective RGB color generation. In Figure 3, the reflection peak doesn’t correspond to the F-P resonance, while reflection valleys correspond to the F-P resonance.”
Comment #4: “The electric field is mentioned in the abstract. The e-field intensity needs to be included in the manuscript.”
Our response: We thank the reviewer for highlight this point. However, the normalized electric field intensity profiles of the green colored structures with the different TiO2 layer thickness are already provided in Figure 5. Thus, we didn’t modify the manuscript.

Round 2
Reviewer 2 Report
Accept as it is
Reviewer 4 Report
I am happy with the revision